# Validation of the COmprehensive Score for Financial Toxicity (COST) in Vietnamese patients with cancer

Binh Thang Tran[1]*, Dinh Duong Le[1], Thanh Gia Nguyen[1], Minh Tu Nguyen[2], Minh Hanh Nguyen[3], Cao Khoa Dang[1], Dinh Trung Tran[1,4]

1 Faculty of Public Health, University of Medicine and Pharmacy, Hue University, Hue City, Thua Thien Hue Province, Vietnam, 2 Undergraduate Training Office, University of Medicine and Pharmacy, Hue University, Hue City, Thua Thien Hue Province, Vietnam, 3 Oncology Centre, Hue Central Hospital, Hue City, Thua Thien Hue Province, Vietnam, 4 Faculty of Public Health, Da Nang University of Medical Technology and Pharmacy, Da Nang City, Vietnam

* tranbinhthang@hueuni.edu.vn, tbthang@huemed-univ.edu.vn

## Abstract

**Data Availability Statement:** All relevant data are within the manuscript and its Supporting Information files.

### Introduction

The COmprehensive Score for Financial Toxicity (COST) has proven to be a reliable tool for quantifying the impact of financial toxicity (FT) in patients with cancer in clinical and public health settings. However, the COST has not yet been validated in Vietnam. Therefore, we aimed to evaluate its reliability and validity among Vietnamese patients with cancer.

### Methods

A cross-sectional study was conducted in a sample of 300 patients with cancer aged 27–95 years (mean: 58.5±11.2) in a tertiary hospital. The COST was translated into Vietnamese and English and adjusted to suit the local culture. Reliability was evaluated using Cronbach's alpha and McDonald's omega coefficients. The construct and convergent validities were also assessed.

### Results

The COST demonstrated good internal consistency and reliability (Cronbach's alpha = 0.913; McDonald's omega = 0.915). The exploratory factor analysis revealed two factors that explained 64.9% of the variance. The adjusted fit indices indicated a good fit of the model ($\chi^2$ (39) = 67.78, p = 0.003; standardized root mean squared residual = 0.042; Tucker–Lewis index = 0.971; comparative fit index = 0.979; root mean square error of approximation = 0.061, 90% confidence interval = 0.035–0084). Higher COST scores were significantly correlated with higher health-related quality of life (EQ-5D-5L utility score: r = 0.21, p = 0.002; EQ VAS: r = 0.28, p < 0.001). Multivariate quantile regression analysis revealed that female sex, rural residence, and unstable job/unemployment were associated with lower COST scores. There was no statistically significant difference in other factors, including clinical factors (types of cancer, staging, and treatment modalities).

**Funding:** This study was supported by Hue University (DHH 2022 – 04–175). The funder played no role in the study design, data collection and analysis, decision to publish, or manuscript preparation.

**Competing interests:** The authors have declared that no competing interests exist.

## Conclusions

The COST is reliable and valid, making it suitable for assessing FT severity in Vietnamese patients with cancer.

## Introduction

There is an increase in cancer incidence both globally and in Vietnam [1–3]. Notably, people with cancer experience a high economic burden and financial impact, such as financial distress [4, 5]. Consequently, this has been associated with poor adherence to treatment and reduced health-related quality of life (HRQoL) [6, 7], which has negatively impacted their family (education, work, and poverty) [8, 9]. The quantitative economic burden of cancer is mostly predominantly characterized as out-of-pocket expenses in literature [10]. However, understanding its impact has only become a major focus in recent years [11], and the term financial toxicity (FT) has recently been adopted by De Souza et al. [11, 12].

FT refers to the negative effect of cancer on an individual's financial situation and its subsequent impact on physical and mental health. Among all instruments for assessing FT, the COmprehensive Score for Financial Toxicity (COST), which was developed and adapted by De Souza et al. (2014), is the most used and is a clinically relevant patient-reported outcome [11, 12]. This self-administered tool comprises 11 items and has demonstrated good psychometric properties in previous studies conducted in high-income countries [13–18]. However, its validity for assessing financial stress in Vietnam remains unexplored. The findings of this study aim to provide robust evidence regarding COST's ability to measure constructs of interest and its potential applications in Vietnam.

## Materials and methods

### Participants

We conducted a cross-sectional validation study of the COST in a single adult oncology center in a tertiary hospital, which is the largest referral hospital in the central and highlands of Vietnam. A sample size of 300 inpatients with the most common cancer was recruited [19], following the rule of thumb suggested by Wolf et al. [20]. This rule recommends including 10 cases per variable in the model for validation. COST contains 11 items; therefore, a sample of 300 patients was considered ideal for robust validation [3].

### Data collection

After obtaining informed written consent from the inpatients, public health students conducted face-to-face interviews using the constructed questionnaires in a private room. All clinical cancer information was obtained from medical records with the assistance of an oncologist. Data were collected between September 1, 2022, and March 31, 2023.

### Instruments and variables

**Demographic and clinical characteristics.** A demographic survey assessed age, sex, education level, marital status, household income, health-risk behaviors (smoking and drinking status), and health insurance benefits. Clinical information included the cancer type, clinical stage, and treatment modality.

**FT.** *Translation and adaptation.* We obtained the COST instrument through forward-backward translation. This process involved two fluent English oncologists and experts in patient-reported outcomes, who were translating the original English version into Vietnamese. The translated version was then back-translated into English by a different translator who was blinded to the original English version. Discrepancies between the original and back-translated versions were resolved through discussions between the translators and the principal investigator to ensure semantic equivalence.

*Instrument description.* The Vietnamese version of the COST retains the core structure of the original scale and comprises 11 elements. It utilizes a 5-point Likert scale ranging from 0 to 4 (see Supporting Information [S1 File] for the COST scale in Vietnamese). Higher COST scores (0–44) indicate better financial well-being, with items 2, 3, 4, 5, 8, 9, and 10 being reverse scored [12, 21].

**HRQoL assessments.** In this study, we used the EQ-5D-5L instruments (comprising the EQ-5D and a visual analog scale (EQ VAS)), a widely used and culturally appropriate scale to measure HRQoL. This scale is particularly valuable because it has been validated with a specific set of Vietnamese values designed to ensure an accurate interpretation of the results within the Vietnamese context [22, 23]. The sets of values for the EQ-5D and EQ VAS were -0.5115 to 1 and 0 to 100, respectively [22, 23].

## Statistical analysis

Continuous variables were summarized as means, standard deviations (SD), medians, and interquartile ranges (IQR). Categorical variables are described using frequencies and percentages.

The reliability of the scale was evaluated using Cronbach's alpha and McDonald's omega coefficients. A coefficient of $\geq 0.7$ generally indicates good internal consistency [24].

The construct validity involves two steps: exploratory factor analysis (EFA) and confirmatory structure analysis (CFA). EFA identified the underlying factors influencing questionnaire responses. The screen plot and varimax rotation determined the number of factors retained [25]. EFA ensured clear and reliable factors through several strategies: (1) removing items with low loadings ($< 0.4$) for minimal contribution; (2) requiring factors to have at least three items to adequately capture the underlying construct; and (3) removing items with high cross-loadings ($> 0.3$) to minimize ambiguity and enhance factor distinction [26]. Bartlett's sphericity test and the Kaiser-Meyer-Olkin (KMO) confirmed the suitability of the EFA [27, 28]. Building on the initial exploration of the factor structure using EFA (two-factor loading was selected), CFA was used to assess the validity of the identified factor structure [27, 29]. Notably, several fit indices (root mean square error of approximation (RMSEA), goodness-of-fit index (GFI), comparative fit index (CFI), and adjusted goodness-of-fit index (AGFI)) indicated good model fit ($p < 0.001$).

Convergent validity was evaluated using the Pearson correlation coefficient between the COST score and HRQoL (EQ-5D and EQ-VAS). The Shapiro–Wilk W test ($p < 0.001$) indicated that the COST scores were not normally distributed due to skewness. Therefore, non-parametric tests were used to analyze the differences in COST scores based on the characteristics. The Mann–Whitney U test was used for comparisons between two groups, and the Kruskal–Wallis test was used for comparisons between three or more groups. Additionally, a multivariate quantile regression analysis (at the 50th percentile) was conducted to investigate the factors associated with the COST score. All independent variables were included for adjustment in the model. Statistical significance was set at $p < 0.05$. Data analysis was performed using IBM SPSS Statistics (version 29.0) and Amos software (version 22.0).

### Ethical approval

The Institutional Biomedical Ethics Committee of the University of Medicine and Pharmacy of Hue University, Vietnam, approved the study (No. H2022/485).

## Results

### Sample characteristics

Of the 330 hospitalized patients approached, 300 (with a participation rate of 90.9%) consented to participate and completed the survey. Table 1 summarizes the characteristics and COST scores of these patients. The participants had a mean age of 58.4 years (SD = 11.2; range, 27–95 years). Males comprised more than half of the group (69.3%), and 68.3% had at least a secondary school education. Biliary/pancreatic cancer was the most common diagnosis (31.6%), and chemotherapy was administered to more than half of the participants (58.9%). The median and mean COST scores were 13.00, IQR (11–18), and 15.1 ± 7.3, respectively.

### Reliability

Cronbach's alpha coefficients for each domain of the COST measure were 0.898 and 0.914, respectively (Table 2). Overall, Cronbach's alpha coefficients and Omega coefficients were 0.913 and 0.915, respectively.

### Validity

**Construct validity.** Two main components were extracted, representing 64.9% of the cumulative variance. The first principal component consisted of seven items ranging from 0.479 to 0.869, and the second consisted of four items ranging from 0.621 to 0.861. Table 3 shows the principal components and factor loadings of COST.

Fig 1 illustrates the principal components and factor loadings of the COST. The fit of the model was evaluated using several indices. The RMSEA value (0.056) was below the recommended threshold of 0.07, indicating a close fit. Furthermore, the GFI (0.955), CFI (>0.9), and AGFI (0.926) exceeded their respective cutoff values of 0.9, further supporting a well-fitting model ($p < 0.001$).

**Convergent validity.** The convergence validity analysis showed a mildly positive correlation between the COST measure and HRQoL. Specifically, the COST scores were positively correlated with the utility score EQ-5D (r = 0.210, p = 0.002) and EQ VAS (r = 0.273, p < 0.0001) (Fig 2). Bivariate analysis revealed several factors that were significantly associated with the COST score (all $p < 0.05$). These factors included female sex, residential area, level of education, occupation, economic status of the household, health insurance copayment, cancer type, and treatment modality (Table 1). In the multivariate model, we found that female sex (coefficient: -5.96, 95% confidence interval (CI): -10.39 –-1.54), residence in rural areas (coefficient: -4.57, 95% CI: -7.41 –-1.73) and occupation with unstable income jobs/unemployment (coefficients: -5.39, 95% CI: -8.71 ––2.07; and -4.64, 95% CI: -8.63 ––0.65, respectively) were statistically associated with a lower COST score. There was no statistical significance for other factors, including clinical factors (cancer type, staging, or treatment modalities) (Table 4).

## Discussion

In this study, we aimed to validate the use of the COST in a Vietnamese population of hospitalized patients with cancer. Our findings provide evidence for the COST's reliability, construct validity, and convergent validity, suggesting its potential as a valuable tool for identifying and measuring FT in this population.

**Table 1. General characteristics of patients with cancer and their COST score (n = 300).**

| Characteristics | N (%) | COST score Mean±SD | Median | IQR (25th -75th) | p value* |
|---|---|---|---|---|---|
| All | 300 | 15.1 (7.3) | 13 | 11–18 | |
| Age | | | | | 0.0747 |
| < 60 | 152 (50.7) | 14.1 (6.2) | 13 | 11–16.5 | |
| ≥ 60 | 148 (49.3) | 16.2 (8.2) | 14 | 11–20.5 | |
| Mean±SD (Max–Min) | 58.5±11.9 | | | | |
| Gender | | | | | 0.0121 |
| Males | 208 (69.3) | 15.8 (7.8) | 14 | 11–20 | |
| Females | 92 (30.7) | 13.4 (6.0) | 13 | 9.5–17 | |
| Education level | | | | | 0.0030 |
| Less than secondary school | 95 (31.7) | 13.5 (6.1) | 12 | 11–15 | |
| From secondary school and above | 205 (68.3) | 15.8 (7.8) | 14 | 11–20 | |
| Living areas | | | | | 0.0031 |
| Urban | 226 (75.3) | 14.3 (6.8) | 13 | 11–17 | |
| Rural | 74 (24.7) | 17.7 (8.3) | 15 | 12–23 | |
| Religious | | | | | 0.288 |
| No | 248 (82.7) | 15.2 (7.5) | 13 | 11–18 | |
| Yes | 52 (17.3) | 14.4 (6.4) | 13 | 11–17.5 | |
| Marital status | | | | | 0.961 |
| Married | 277 (92.3) | 15.1 (7.4) | 13 | 11–18 | |
| Single/Widow/Divorced | 23 (7.7) | 14.7 (7.1) | 14 | 9–19 | |
| Occupation | | | | | 0.0001 |
| Stable income jobs | 66 (22.0) | 18.8 (9.2) | 16.5 | 13–23 | |
| Unstable income jobs | 163 (54.3) | 13.5 (6.5) | 13 | 10–16 | |
| No income | 71 (23.7) | 15.4 (6) | 14 | 11–19 | |
| Economic status | | | | | 0.0070 |
| Poor | 19 (6.3) | 11.5 (6.0) | 11 | 8–13 | |
| Near poor | 24 (8.0) | 13.1 (6.2) | 12 | 11–15.5 | |
| Average | 257 (85.7) | 15.6 (7.4) | 13 | 11–19 | |
| Health insurance benefits | | | | | 0.5507 |
| 100% | 153 (51.0) | 15.4 (8.1) | 13 | 11–19 | |
| 95% | 36 (12.0) | 15.9 (6.6) | 14 | 11.5–20.5 | |
| 80% | 111 (37.0) | 14.4 (6.5) | 13 | 11–17 | |
| Type of cancers | | | | | 0.0252 |
| Oesophagus cancer | 39 (13.0) | 12.3 (6.0) | 12 | 8–15 | |
| Stomach cancer | 15 (5.0) | 13.3 (3.3) | 13 | 13–16 | |
| Colorectal cancer | 30 (10.0) | 15.9 (7.2) | 13 | 11–22 | |
| Liver cancer | 32 (10.7) | 15.4 (7.1) | 15 | 10–18.5 | |
| Biliary/pancreatic cancer | 112 (37.3) | 16.6 (8.3) | 14 | 11–21 | |
| Breast cancer | 72 (24.0) | 14.2 (6.8) | 13 | 11–15 | |
| Stages of cancer | | | | | 0.343 |
| I | 6 (2.0) | 12.3 (2.8) | 11.5 | 11–14 | |
| II | 54 (18.0) | 13.8 (7.8) | 13 | 9–18 | |
| III | 110 (36.7) | 15.2 (7.1) | 13 | 11–16 | |
| IV | 130 (43.3) | 15.7 (7.4) | 13 | 11–20 | |
| Treatment modality | | | | | 0.0385 |
| Surgery | 4 (1.3) | 18.5 (17.4) | 12.5 | 8.5–28.5 | |

*(Continued)*

**Table 1.** (Continued)

| Characteristics | N (%) | COST score Mean±SD | Median | IQR (25th -75th) | p value* |
|---|---|---|---|---|---|
| Surgery and chemotherapy | 153 (51.0) | 16.3 (8.0) | 14 | 11–20 | |
| Chemotherapy | 83 (27.7) | 13.2 (5.8) | 13 | 10–16 | |
| Radiotherapy | 54 (18.0) | 15 (6.3) | 13 | 11–17 | |
| Palliative care | 6 (2.0) | 10.5 (4.0) | 10.5 | 8–14 | |
| Alcohol use | | | | | 0.670 |
| Yes | 175 (58.3) | 14.9 (7.3) | 13 | 11–17 | |
| No | 125 (41.7) | 15.4 (7.4) | 13 | 11–19 | |
| Cigarette status | | | | 11–17 | 0.953 |
| Yes | 146 (48.7) | 14.9 (6.8) | 13 | 10–19 | |
| No | 154 (51.3) | 15.3 (7.8) | 13 | 11–17 | |

*$p$ value was calculated using non-parametric tests (Mann-Whitney U test or Kruskal-Wallis). Alcohol use and cigarette status were asked about their current situation. IQR, Interquartile range.

## Reliability

Cronbach's alpha coefficients for individual domains (0.8976 and 0.9137) and the general scale (0.913) exceeded the commonly accepted threshold of 0.7, indicating high internal consistency. Furthermore, Omega's coefficient ($\omega = 0.915$) confirmed high reliability. These findings are consistent with previous studies that have reported a good reliability (Cronbach's alpha) of COST in different populations such as Brazil, Japan, and India at 0.83, 0.87, and 0.92, respectively [16–18].

## Validity

This finding aligns with the theoretical framework of COST and supports its validity. EFA identified two distinct factors that accounted for 64.9% of the cumulative variance. All items

**Table 2. Reliability of COST.**

| Item code | Details of each item | Item-test correlation | Item-rest correlation | Internal Consistency Cronbach's α |
|---|---|---|---|---|
| FT1 | I know that I have enough money in savings, retirement, or assets to cover the costs of my treatment | 0.6897 | 0.614 | 0.908 |
| FT2 | My out-of-pocket medical expenses are more than I thought they would be | 0.606 | 0.532 | 0.911 |
| FT3 | I worry about the financial problems I will have in the future as a result of my illness or treatment | 0.837 | 0.791 | 0.898 |
| FT4 | I feel I have no choice about the amount of money I spend on care | 0.600 | 0.536 | 0.911 |
| FT5 | I am frustrated that I cannot work or contribute as much as I usually do | 0.630 | 0.530 | 0.914 |
| FT6 | I am satisfied with my current financial situation | 0.807 | 0.755 | 0.900 |
| FT7 | I am able to meet my monthly expenses | 0.679 | 0.600 | 0.909 |
| FT8 | I feel financially stressed | 0.847 | 0.803 | 0.898 |
| FT9 | I am concerned about keeping my job and income, including paid work at home | 0.791 | 0.735 | 0.902 |
| FT10 | My cancer or treatment has reduced my satisfaction with my present financial situation | 0.790 | 0.745 | 0.902 |
| FT11 | I feel in control of my financial situation | 0.782 | 0.720 | 0.902 |
| Cronbach's alpha | | | | 0.913 |
| McDonald's Omega coefficients | | | | 0.915 |

**Table 3. EFA of the final 11-item COST measure with 2-factors solution.**

| Item code | Details of each item | Factor 1 | Factor 2 |
|---|---|---|---|
| FT9 | I am concerned about keeping my job and income, including paid work at home | 0.869 | |
| FT8 | I feel financially stressed | 0.768 | |
| FT3 | I worry about the financial problems I will have in the future as a result of my illness or treatment | 0.756 | |
| FT5 | I am frustrated that I cannot work or contribute as much as I usually do | 0.699 | |
| FT4 | I feel I have no choice about the amount of money I spend on care | 0.643 | |
| FT10 | My cancer or treatment has reduced my satisfaction with my present financial situation | 0.611 | |
| FT2 | My out-of-pocket medical expenses are more than I thought they would be | 0.479 | |
| FT7 | I am able to meet my monthly expenses | | 0.861 |
| FT11 | I feel in control of my financial situation | | 0.855 |
| FT1 | I know that I have enough money in savings, retirement, or assets to cover the costs of my treatment | | 0.690 |
| FT6 | I am satisfied with my current financial situation | | 0.621 |

Extraction Method: Principal Axis Factoring. Rotation Method: Promax with Kaiser Normalization. a. Rotation converged in 3 iterations. EFA, Exploratory factor analysis.

were loaded into these factors with loads ranging from 0.479 to 0.869. The first factor included elements FT2, FT3, FT4, FT5, FT8, FT9, and FT10, whereas the second factor included elements FT1, FT6, FT7, and FT11.

Our results are consistent with those reported in China by Yu et al. (2021), who found a cumulative variance of 63.04%. Similar to our study, their analysis revealed two factors, with the first factor containing seven elements and the second containing four. Yu et al. suggested that the COST scale could be conceptualized as measuring two dimensions: positive wealth status and negative psychosocial aspects [15]. However, a study by Shim et al. (2022) in Korea also identified a two-factor structure; however, this study had some differences in item loadings. The first factor included items FT3, FT5, FT8, FT9, and FT10, and the second factor included items FT1, FT4, FT6, and FT7. Items FT4, FT11 (which appeared in both factors in their analysis but with a low load), and FT2 were not included in either factor. The authors attributed this discrepancy to linguistic validation issues; however, they concluded that the general validity of the scale remained good [13].

Our findings differ from those of the original development of the COST scale, which reported a one-factor structure explaining 93% of the variance in an English-speaking sample. Other studies have also reported variations. Dar et al. (2021) identified a one-factor structure that explained 56% of the variance in India [16]. However, Sharif et al. (2021) identified a three-factor structure that explained 65% of the variance in Iran [14]. These discrepancies highlight the importance of COST validation in various contexts. However, in our data analysis, achieving an explained variance of over 80% required five-factor loadings, which is not ideal for the COST scale with only 11 items. Our final two-factor model demonstrated acceptable fit indices consistent with the findings of Yu et al. in China [15]. These comparisons suggest that the COST is likely to measure distinct but related constructs relevant to FT in patients with cancer.

## Convergent validity

The significant positive correlation between the COST and HRQoL measured using the EQ-5D or EQ VAS supports the convergent validity of the COST. This finding provides evidence

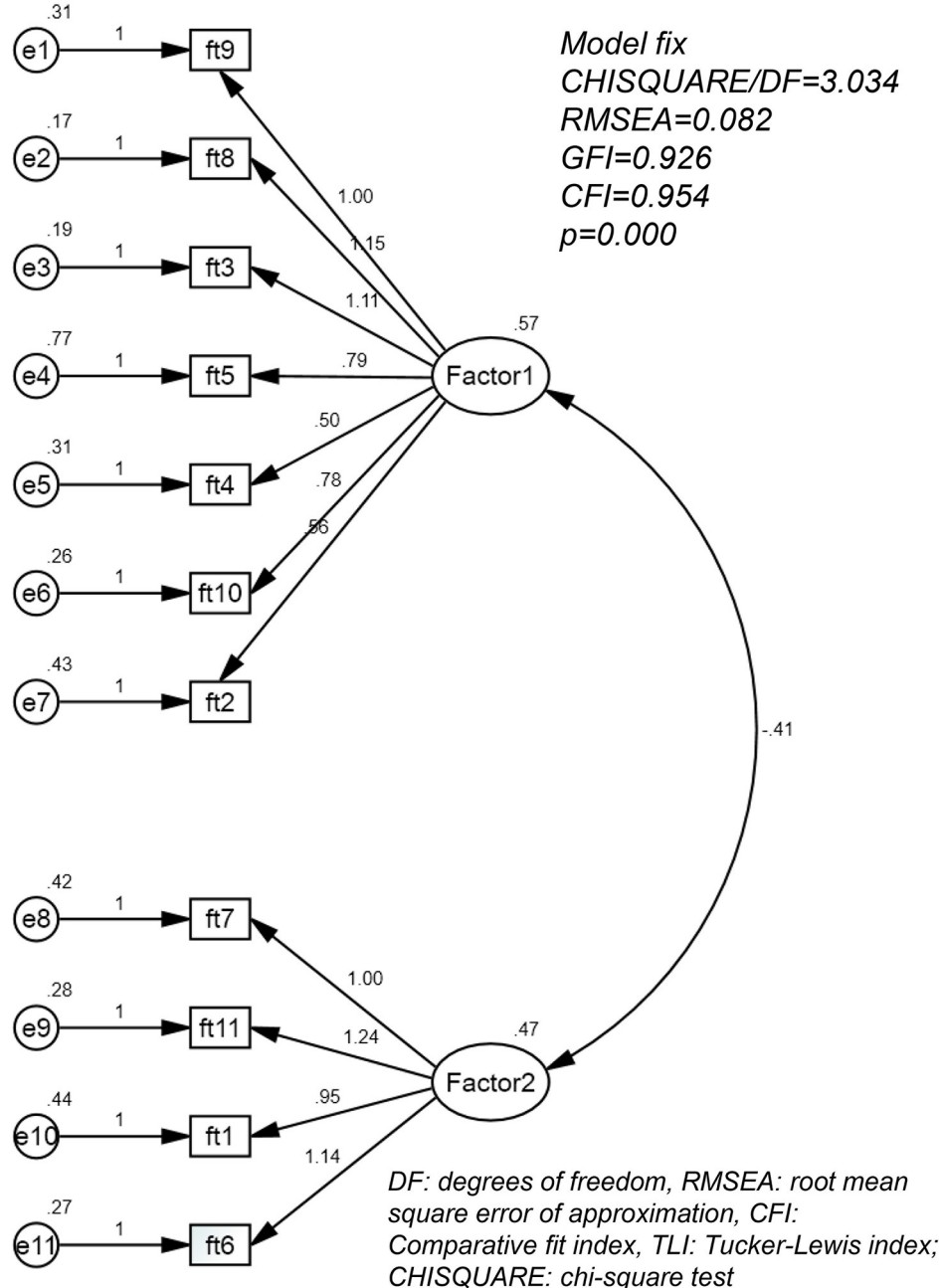

**Fig 1. Structure of items in the COST.**

that patients with higher COST scores experience higher HRQoL; consequently, a higher level of FT is associated with worse HRQoL, which is consistent with previous studies [7, 30, 31]. In addition, factors such as low socioeconomic status and cancer treatment bolster evidence for the multidimensionality of the FT [6, 32] and the validity of the COST for application across diverse clinical characteristics and settings [7].

To our knowledge, this is the first study to validate the COST in patients with cancer in Vietnam. Second, the sample size for this study was sufficient, particularly for scale validation, which suggests a range of 30–460 [20].

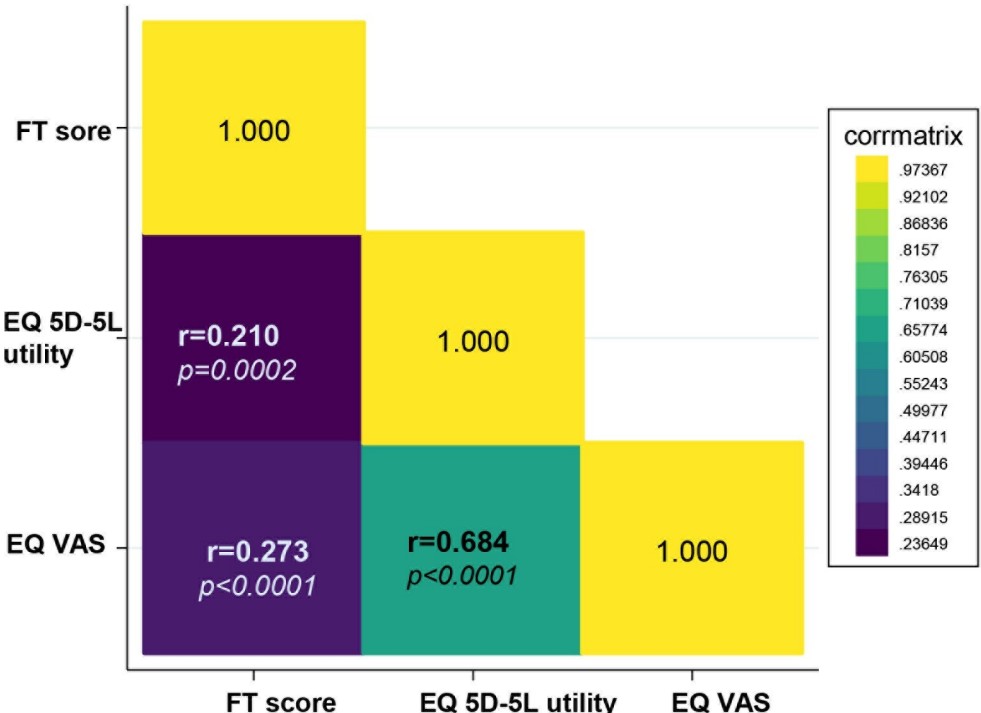

**Fig 2. Heat map of the Peason correlation coefficient matrix for COST score and HQOL (EQ 5D and EQ VAS).**

**Table 4. A multivariate quantile regression analysis for predictors of COST score.**

| Factors | Coefficient | [95% CI] | p-value |
|---|---|---|---|
| Age group ($\geq 60$ vs. <60) | 2.50 | -0.12 to 5.12 | 0.062 |
| Gender (*Females vs. Males*) | -5.96 | -10.39 to -1.54 | 0.008 |
| Education level (*From secondary school and above vs. Less than secondary school*) | 0.39 | -2.46 to 3.24 | 0.786 |
| Residence areas (*Rural vs. Urban*) | -4.57 | -7.41 to 1.73 | 0.002 |
| Religious (*Yes vs. No*) | -1.50 | -4.74 to 1.74 | 0.363 |
| Marital status (*Yes vs. No*) | 1.11 | -3.44 to 5.65 | 0.632 |
| Occupation (*vs. Stable income jobs*) | | | |
| Unstable income jobs | -5.39 | -8.71 to -2.07 | 0.002 |
| No income | -4.64 | -8.63 to -0.65 | 0.023 |
| Family economic status (*vs. Poor*) | | | |
| Near poor | 0.82 | -5.58 to 7.23 | 0.801 |
| Average | 2.96 | -2.1 to 8.03 | 0.250 |
| Health insurance copayment (*vs. 100%*) | | | |
| 95% | -4.07 | -8.35 to 0.2 | 0.062 |
| 80% | -0.86 | -3.49 to 1.78 | 0.522 |
| Types of cancer (*vs. Oesophagus cancer*) | | | |
| Stomach cancer | 1.29 | -6.64 to 9.21 | 0.750 |
| Colorectal cancer | 2.68 | -3.32 to 8.68 | 0.380 |
| Liver cancer | 3.00 | -2.39 to 8.39 | 0.274 |
| Biliary/pancreatic cancer | 2.04 | -2.77 to 6.84 | 0.405 |

*(Continued)*

**Table 4.** (Continued)

| Factors | Coefficient | [95% CI] | p-value |
|---|---|---|---|
| Breast cancer | 0.07 | -5.66 to 5.8 | 0.980 |
| Stages of cancer (*vs. Stage I*) | | | |
| II | 4.18 | -4.63 to 12.99 | 0.351 |
| III | 4.36 | -4.19 to 12.91 | 0.317 |
| IV | 5.11 | -3.52 to 13.74 | 0.245 |
| Treatment modality (*vs. Surgery*) | | | |
| Surgery and chemotherapy | 1.68 | -8.87 to 12.23 | 0.754 |
| Chemotherapy | -0.25 | -10.8 to 10.3 | 0.963 |
| Radiotherapy | 0.11 | -10.85 to 11.06 | 0.985 |
| Palliative care | -5.54 | -18.94 to 7.87 | 0.417 |
| Alcohol use (*No vs. Yes*) | 2.86 | -0.64 to 6.36 | 0.109 |
| Cigarette smoking status (*No vs. Yes*) | 2.36 | -0.98 to 5.69 | 0.165 |

CI, Confidence Interval

This study has some limitations. First, the cross-sectional design limited our ability to evaluate how the FT changes over time during treatment. Previous research in China has shown fluctuations in FT throughout treatments [15]. Second, our sample included mainly patients with common cancers, most of whom were in advanced stages. This limits the generalizability of our findings to other types and specific stages of cancer. Third, we did not establish a threshold score for the COST scale because of the lack of a gold standard for measuring FT. Future research should investigate possible threshold scores to identify patients who may benefit from FT-targeted interventions.

## Conclusions

Our study provides positive preliminary evidence for the reliability, construct, and convergent validity of the COST in a Vietnamese population of hospitalized patients with cancer. This suggests that the COST could be a valuable tool for healthcare professionals to assess and monitor FT in this population. Future research with larger and more diverse samples is required to confirm these findings and explore the potential applications of COST in clinical and public health settings.

## Supporting information

**S1 File. Vietnamese version of COST.**
(DOCX)

**S1 Data.**
(CSV)

## Author Contributions

**Conceptualization:** Binh Thang Tran, Dinh Duong Le, Thanh Gia Nguyen, Minh Tu Nguyen, Minh Hanh Nguyen, Cao Khoa Dang.

**Data curation:** Binh Thang Tran, Dinh Duong Le, Thanh Gia Nguyen, Minh Tu Nguyen, Minh Hanh Nguyen, Cao Khoa Dang.

**Formal analysis:** Binh Thang Tran, Dinh Duong Le, Thanh Gia Nguyen, Minh Tu Nguyen, Minh Hanh Nguyen, Cao Khoa Dang, Dinh Trung Tran.

**Funding acquisition:** Binh Thang Tran, Thanh Gia Nguyen, Minh Tu Nguyen.

**Investigation:** Binh Thang Tran, Dinh Duong Le, Thanh Gia Nguyen, Minh Tu Nguyen, Minh Hanh Nguyen, Cao Khoa Dang, Dinh Trung Tran.

**Methodology:** Binh Thang Tran, Dinh Duong Le, Thanh Gia Nguyen, Minh Tu Nguyen, Minh Hanh Nguyen, Cao Khoa Dang, Dinh Trung Tran.

**Project administration:** Binh Thang Tran, Minh Tu Nguyen.

**Resources:** Binh Thang Tran, Dinh Trung Tran.

**Software:** Binh Thang Tran, Dinh Duong Le, Dinh Trung Tran.

**Supervision:** Binh Thang Tran, Dinh Trung Tran.

**Validation:** Binh Thang Tran.

**Visualization:** Binh Thang Tran.

**Writing – original draft:** Binh Thang Tran.

**Writing – review & editing:** Binh Thang Tran.

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
