## [Decision Letter · Decision Letter 0]

15 May 2024

PONE-D-24-12510Validation of the Vietnamese Comprehensive Score for Financial Toxicity (COST) in Cancer PatientsPLOS ONE

Dear Dr. Tran,

Thank you for submitting your manuscript to PLOS ONE. After careful consideration, we feel that it has merit but does not fully meet PLOS ONE’s publication criteria as it currently stands. Therefore, we invite you to submit a revised version of the manuscript that addresses the points raised during the review process.

We look forward to receiving your revised manuscript.

Kind regards,

Le An Pham, Ph.D,MD

Academic Editor

PLOS ONE

Journal Requirements:

Reviewers' comments:

Reviewer's Responses to Questions

**Comments to the Author**

1. Is the manuscript technically sound, and do the data support the conclusions?

Reviewer #1: Yes

Reviewer #2: Yes

2. Has the statistical analysis been performed appropriately and rigorously? 

Reviewer #1: Yes

Reviewer #2: No

3. Have the authors made all data underlying the findings in their manuscript fully available?

Reviewer #1: Yes

Reviewer #2: No

4. Is the manuscript presented in an intelligible fashion and written in standard English?

Reviewer #1: No

Reviewer #2: Yes

5. Review Comments to the Author

Reviewer #1: The paper offers a comprehensive validation study of the Vietnamese version of the Comprehensive Score for Financial Toxicity (COST) instrument among cancer patients. While the methodology is well-described and the study design seems robust, there are formatting inconsistencies and minor issues that require attention before publication.

1. Throughout the manuscript, there are multiple variations of the term for the Vietnamese version of COST. It is crucial to use only one consistent term for clarity. Please ensure that the same term is used consistently across the entire manuscript.

2. The paragraph format in the statistical analysis section lacks consistency. It is suggested to remove the headings "Descriptive statistics" and "Internal consistency" to maintain coherence.

3. Recheck the format of references and citations to ensure consistency and accuracy. Pay attention to punctuation, capitalization, and formatting style (e.g., APA, MLA) to adhere to journal guidelines.

4. The term "Exploratory factor analysis" is redundant as it has already been explained previously. Consider removing it for clarity and conciseness.

5. Additional Comments: Please double-check for any formatting errors or inconsistencies throughout the manuscript to maintain readability.

Reviewer #2: The study “Validation of the Vietnamese Comprehensive Score for Financial Toxicity (COST) in Cancer Patients” was performed by Tran et al.

The paper is well-written and showed that the COmprehensive Score for financial Toxicity (COST) demonstrated good internal consistency and The Vietnamese COST is reliable and valid.

There were several unclear issues and limitations that need to be addressed:

- For the principal component analysis, two factors explained more than 64% of the variance, that a low explanation. With this analysis, you can consider adding more factors to have more than 80% of the variance.

- Why did the authors express the COST score by both Mean ± SD and Median? This should be based on the distribution of COST score in each group, normal or non-normal distribution. Please also verify the statistical analysis, which tests were used to analyze the differences of COST score: t-test, ANOVA, Kruskal-Wallis or Mann Whitney?

- A linear regression should be performed to understand the contribution of independent variables to the COST score.

6. PLOS authors have the option to publish the peer review history of their article (what does this mean?). If published, this will include your full peer review and any attached files.

Reviewer #1: No

Reviewer #2: **Yes: **Minh Duc Do

---

## [Author Response · Author response to Decision Letter 0]

20 May 2024

Date: May 20th, 2024

Dear Editor and Reviewers

We would like to thank the editors and reviewers for their constructive comments and suggestions on our paper. We appreciate their feedback and have incorporated substantial revisions in our revised version. All changes have been highlighted in yellow. Once again, thank you very much. 

Journal requirement

Journal Requirements: When submitting your revision, we need you to address these additional requirements.

Reply: We have addressed all the issues raised and made corrections to ensure compliance with journal requirements.

The title of study

All: We have modified the title of study as “Validation of the COmprehensive Score for Financial Toxicity (COST) in Vietnamese Patients with Cancer” 

Reviewer 1

Reviewer #1: The paper offers a comprehensive validation study of the Vietnamese version of the Comprehensive Score for Financial Toxicity (COST) instrument among cancer patients. While the methodology is well-described and the study design seems robust, there are formatting inconsistencies and minor issues that require attention before publication.

1. Throughout the manuscript, there are multiple variations of the term for the Vietnamese version of COST. It is crucial to use only one consistent term for clarity. Please ensure that the same term is used consistently across the entire manuscript.

Reply 1: We have corrected all the terms used for COST, Financial toxictiy and others statistical analysis. 

2. The paragraph format in the statistical analysis section lacks consistency. It is suggested to remove the headings "Descriptive statistics" and "Internal consistency" to maintain coherence.

Reply 2: We have modified this part for consistency (page 5-6)

Statistical analysis

Continuous variables were summarized using means, standard deviations (SD), median, and Interquartile range (IQR). Categorical variables were described using frequencies and percentages. 

The reliability of the scale was assessed using Cronbach's alpha and McDonald's omega coefficients. A coefficient of 0.7 or higher generally indicates good internal consistency [24].

Construct validity involved two steps including exploratory factor analysis (EFA) and confirmatory structure analysis (CFA). EFA identified underlying factors influencing questionnaire responses. The scree plot and varimax rotation determined the number of factors retained [25]. EFA ensured clear and reliable factors through several strategies: (1) Removing items with low loadings (< 0.4) for minimal contribution; (2) requiring factors to have at least 3 items to capture the underlying construct adequately; (3) Removing items with high cross-loadings (> 0.3) to minimize ambiguity and enhance factor distinction [26]. Bartlett's sphericity test and Kaiser-Meyer-Olkin (KMO) confirmed the suitability of EFA [27, 28]. Building on the initial exploration of the factor structure using EFA (two factors loading was selected), CFA was then employed to assess the validity of the identified factor structure [27, 29]. Several fit indices (Root Mean Square Error of Approximation (RMSEA), Goodness-of-Fit Index (GFI), Comparative Fit Index (CFI), and Adjusted Goodness-of-Fit Index (AGFI)) indicated a good model fit (p < 0.001). 

Convergent validity was assessed using the Pearson correlation coefficient between the COST score and HRQoL (EQ-5D and EQ VAS). Shapiro-Wilk W test (p-value < 0.001) indicated that the COST score were not normally distributed due to skewness. Therefore, nonparametric tests were used to analyze the differences in COST score by characteristics. The Mann-Whitney U test was used for comparisons between two groups, and the Kruskal-Wallis test was used for comparisons between three or more groups. Additionally, a multivariate quantile regression analysis (at the 50th percentile) was conducted to investigate factors associated with the COST score. All independent variables were included for adjustment in the model. The threshold of a statistically significant level was less than 0.05. Data analysis was performed with Stata software (version 13.4.2) and Amos software (version 22.0).

3. Recheck the format of references and citations to ensure consistency and accuracy. Pay attention to punctuation, capitalization, and formatting style (e.g., APA, MLA) to adhere to journal guidelines.

Reply 3: We have checked and corrected all citations and references in compliance with journal requirement. 

4. The term "Exploratory factor analysis" is redundant as it has already been explained previously. Consider removing it for clarity and conciseness.

Reply 4: Thank you for your thorough review. We have removed duplicate information from the statistical analysis and revised the entire section accordingly. (see reply 2)

5. Additional Comments: Please double-check for any formatting errors or inconsistencies throughout the manuscript to maintain readability.

Reply 5: Thank you again for your suggestion. We have carefully reviewed the document and made substantial revisions.

Reviewer 2

Reviewer #2: The study “Validation of the Vietnamese Comprehensive Score for Financial Toxicity (COST) in Cancer Patients” was performed by Tran et al.

The paper is well-written and showed that the COmprehensive Score for financial Toxicity (COST) demonstrated good internal consistency and The Vietnamese COST is reliable and valid.

There were several unclear issues and limitations that need to be addressed:

6. For the principal component analysis, two factors explained more than 64% of the variance, that a low explanation. With this analysis, you can consider adding more factors to have more than 80% of the variance.

Reply 6: Thank you for raising a critical point regarding the validation steps. After carefully checking our data and re-analyzing it, we found that achieving over 80% variance requires at least five factors to load. Unfortunately, this is not ideal for our scale with only 11 items. We have addressed this limitation in the discussion section on page 10 to further explain our results. (Page 10) 

Although, in our data analysis, achieving an explained variance of over 80% requires five factor loadings, which is not ideal for the COST scale with only 11 items. Our final two-factor model demonstrates acceptable fit indices, consistent with the findings of Yu et al in China [15]. These comparisons suggest that the COST likely measures distinct but related constructs relevant to FT in cancer patients. 

We have coppied output from AMOS for reference 

Total Variance Explained

Component Initial Eigenvalues Extraction Sums of Squared Loadings Rotation Sums of Squared Loadings

 Total % of Variance Cumulative % Total % of Variance Cumulative % Total % of Variance Cumulative %

1 6.003 54.572 54.572 6.003 54.572 54.572 3.747 34.067 34.067

2 1.135 10.317 64.889 1.135 10.317 64.889 3.390 30.822 64.889

3 .724 6.582 71.471 

4 .645 5.863 77.334 

5 .566 5.145 82.479 

6 .486 4.415 86.894 

7 .382 3.472 90.366 

8 .346 3.148 93.514 

9 .297 2.704 96.218 

10 .269 2.443 98.661 

11 .147 1.339 100.000 

Extraction Method: Principal Component Analysis.

7. Why did the authors express the COST score by both Mean ± SD and Median? This should be based on the distribution of COST score in each group, normal or non-normal distribution. Please also verify the statistical analysis, which tests were used to analyze the differences of COST score: t-test, ANOVA, Kruskal-Wallis or Mann Whitney?

Reply 7: Thank you for identifying the issues in our statistical analysis. We have addressed these concerns by revising the statistical method and incorporating the median, IQR (interquartile range), and p-value into Table 1.

Statistical method (Page 6):

Convergent validity was assessed using the Pearson correlation coefficient between the COST score and HRQoL (EQ-5D and EQ VAS). Shapiro-Wilk W test (p-value < 0.001) indicated that the COST scores were not normally distributed due to skewness. Therefore, nonparametric tests were used to analyze the differences in COST score by characteristics. The Mann-Whitney U test was used for comparisons between two groups, and the Kruskal-Wallis test was used for comparisons between three or more groups. Additionally, a multivariate quantile regression analysis (at the 50th percentile) was conducted to investigate factors associated with the COST score. All independent variables were included for adjustment in the model. The threshold of a statistically significant level was less than 0.05. Data analysis was performed with Stata software (version 13.4.2) and Amos software (version 22.0).

Table 1:

Chracteristics N (%) COST score Mean±SD Median IQR (25th -75th ) p value*

All 300 15.1 (7.3) 13 11-18 

Age 0.0747

< 60 152 (50.7) 14.1 (6.2) 13 11-16.5 

≥ 60 148 (49.3) 16.2 (8.2) 14 11-20.5 

Mean±SD

(Max – Min) 58.5±11.9 

Sex 0.0121 

Men 208 (69.3) 15.8 (7.8) 14 11-20 

Women 92 (30.7) 13.4 (6.0) 13 9.5-17 

Education level 0.0030 

Less than secondary school 95 (31.7) 13.5 (6.1) 12 11-15 

From secondary school and above 205 (68.3) 15.8 (7.8) 14 11-20 

Living areas 0.0031 

Urban 226 (75.3) 14.3 (6.8) 13 11-17 

Rural 74 (24.7) 17.7 (8.3) 15 12-23 

Religious 0.288

No 248 (82.7) 15.2 (7.5) 13 11-18 

Yes 52 (17.3) 14.4 (6.4) 13 11-17.5 

Marital status 0.961

Married 277 (92.3) 15.1 (7.4) 13 11-18 

Single/Widow/Dervoce 23 (7.7) 14.7 (7.1) 14 9-19 

Occupation 0.0001

Stable income jobs 66 (22.0) 18.8 (9.2) 16.5 13-23 

Unstable income jobs 163 (54.3) 13.5 (6.5) 13 10-16 

No income 71 (23.7) 15.4 (6) 14 11-19 

Economic status 0.0070

Poor 19 (6.3) 11.5 (6.0) 11 8-13 

Near poor 24 (8.0) 13.1 (6.2) 12 11-15.5 

Average 257 (85.7) 15.6 (7.4) 13 11-19 

Health insurance benefits 0.5507

100% 153 (51.0) 15.4 (8.1) 13 11-19 

95% 36 (12.0) 15.9 (6.6) 14 11.5-20.5 

80% 111 (37.0) 14.4 (6.5) 13 11-17 

Type of cancers 0.0252

Oesophagus cancer 39 (13.0) 12.3 (6.0) 12 8-15 

Stomach cancer 15 (5.0) 13.3 (3.3) 13 13-16 

Colorectal cancer 30 (10.0) 15.9 (7.2) 13 11-22 

Liver cancer 32 (10.7) 15.4 (7.1) 15 10-18.5 

Biliary/pancreatic cancer 112 (37.3) 16.6 (8.3) 14 11-21 

Breast cancer 72 (24.0) 14.2 (6.8) 13 11-15 

Stages of cancer 0.343

I 6 (2.0) 12.3 (2.8) 11.5 11-14 

II 54 (18.0) 13.8 (7.8) 13 9-18 

III 110 (36.7) 15.2 (7.1) 13 11-16 

IV 130 (43.3) 15.7 (7.4) 13 11-20 

Treatment modality 0.0385

Surgery 4 (1.3) 18.5 (17.4) 12.5 8.5-28.5 

Surgery and chemotherapy 153 (51.0) 16.3 (8.0) 14 11-20 

Chemotherapy 83 (27.7) 13.2 (5.8) 13 10-16 

Radiotherapy 54 (18.0) 15 (6.3) 13 11-17 

Palliative care 6 (2.0) 10.5 (4.0) 10.5 8-14 

Alcohol use 0.670

Yes 175 (58.3) 14.9 (7.3) 13 11-17 

No 125 (41.7) 15.4 (7.4) 13 11-19 

Cigarette status 11-17 0.953

Yes 146 (48.7) 14.9 (6.8) 13 10-19 

No 154 (51.3) 15.3 (7.8) 13 11-17 

8. A linear regression should be performed to understand the contribution of independent variables to the COST score.

Reply 8: We appreciate your suggestion once again. To address the skewed distribution of the COST score, we performed a multivariate quantile regression model to identify its predictors. We focused on the 50th percentile (median) of the COST score in this regression analysis. All results are presented in Table 4 and the results section (Page 8). 

Table 4. A multivariate quantile regression analysis for predictors of COST score.

Factors Coefficient [95% CI] p-value

Age group (≥ 60 vs. <60) 2.50 -0.12 to 5.12 0.062

Sex (Women vs. Men) -5.96 -10.39 to -1.54 0.008

Education level (From secondary school and above vs.Less than secondary school) 0.39 -2.46 to 3.24 0.786

Residence areas (Rural vs. Urban ) -4.57 -7.41 to 1.73 0.002

Religious (Yes vs. No) -1.50 -4.74 to 1.74 0.363

Marital status (Yes vs. No) 1.11 -3.44 to 5.65 0.632

Occupation (vs. Stable income jobs) 

Unstable income jobs -5.39 -8.71 to -2.07 0.002

No income -4.64 -8.63 to -0.65 0.023

Family economic status (vs. Poor) 

Near poor 0.82 -5.58 to 7.23 0.801

Average 2.96 -2.1 to 8.03 0.250

Health insurance copayment (vs. 100%) 

95% -4.07 -8.35 to 0.2 0.062

80% -0.86 -3.49 to 1.78 0.522

Types of cancer (vs. Oesophagus cancer) 

Stomach cancer 1.29 -6.64 to 9.21 0.750

Colorectal cancer 2.68 -3.32 to 8.68 0.380

Liver cancer 3.00 -2.39 to 8.39 0.274

Biliary/pancreatic cancer 2.04 -2.77 to 6.84 0.405

Breast cancer 0.07 -5.66 to 5.8 0.980

Stages of cancer (vs. Stage I) 

II 4.18 -4.63 to 12.99 0.351

III 4.36 -4.19 to 12.91 0.317

IV 5.11 -3.52 to 13.74 0.245

Treatment modality (vs. Surgery) 

Surgery and chemotherapy 1.68 -8.87 to 12.23 0.754

Chemotherapy -0.25 -10.8 to 10.3 0.963

Radiotherapy 0.11 -10.85 to 11.06 0.985

Palliative care -5.54 -18.94 to 7.87 0.417

Alcohol use (No vs. Yes) 2.86 -0.64 to 6.36 0.109

Cigarette smoking status (No vs. Yes) 2.36 -0.98 to 5.69 0.165

CI, Confidence Interval

---

## [Editor Report · Decision Letter 1]

27 May 2024

PONE-D-24-12510R1Validation of the COmprehensive Score for Financial Toxicity (COST) in Vietnamese Patients with CancerPLOS ONE

Dear Dr. Tran,

Thank you for submitting your manuscript to PLOS ONE. After careful consideration, we feel that it has merit but does not fully meet PLOS ONE’s publication criteria as it currently stands. Therefore, we invite you to submit a revised version of the manuscript that addresses the points raised during the review process.

We look forward to receiving your revised manuscript.

Kind regards,

Le An Pham, Ph.D,MD

Academic Editor

PLOS ONE

Additional Editor Comments:

Please provide a detailed explanation of the revisions made to the document based on the opinions of two reviewers, using line by line format.

The reason for utilizing both Stata and Amos in the analysis of exploratory factor analysis (EFA) and confirmatory factor analysis (CFA) is to leverage the unique strengths and capabilities of each software tool. I am confident that STATA has the capability to perform these tasks.

Do you believe that the presence of a patient aged 90-95 with an unusual value can significantly affect your results?

Please find some english certificate group that can assist you in preparing good english manuscript

---

## [Author Response · Author response to Decision Letter 1]

7 Jun 2024

Date: June 7th, 2024

Dear Editor and Reviewers

Thank you for your additional comments on our first-round revision. We have incorporated your feedback by merging it with the changes we made in the first round of revisions.

Response to comments on May 27th, 2024 – Round 2

Additional Editor Comments:

2.1. Please provide a detailed explanation of the revisions made to the document based on the opinions of two reviewers, using line by line format.

Reply 2.1: Thank you for your suggestion. We have added in the second part of this response letter and added page and line format

2. The reason for utilizing both Stata and Amos in the analysis of exploratory factor analysis (EFA) and confirmatory factor analysis (CFA) is to leverage the unique strengths and capabilities of each software tool. I am confident that STATA has the capability to perform these tasks.

Reply 2.2: Thank you again for your continued interest in our data analysis. While Stata was initially used for descriptive and multivariate analyses, we have reanalyzed the data using SPSS for consistency, as AMOS is also an SPSS extension. This information has been added on page 7, lines 130-131.

3. Do you believe that the presence of a patient aged 90-95 with an unusual value can significantly affect your results?

Reply 2.3: We have reviewed the data on age and found that the group of patients aged 80-95 represents only 1.67% of the total sample (result below). Therefore, we believe this distribution is unlikely to significantly influence our overall data analysis.

 Age Freq. Percent Cum.

 27 1 0.33 0.33

 32 2 0.67 1.00

 33 1 0.33 1.33

 34 2 0.67 2.00

 35 2 0.67 2.67

 36 1 0.33 3.00

 38 5 1.67 4.67

 39 5 1.67 6.33

 40 7 2.33 8.67

 41 1 0.33 9.00

 42 6 2.00 11.00

 43 3 1.00 12.00

 44 2 0.67 12.67

 45 3 1.00 13.67

 46 4 1.33 15.00

 47 6 2.00 17.00

 48 4 1.33 18.33

 49 7 2.33 20.67

 50 9 3.00 23.67

 51 5 1.67 25.33

 52 6 2.00 27.33

 53 8 2.67 30.00

 54 9 3.00 33.00

 55 11 3.67 36.67

 56 10 3.33 40.00

 57 13 4.33 44.33

 58 12 4.00 48.33

 59 7 2.33 50.67

 60 8 2.67 53.33

 61 10 3.33 56.67

 62 12 4.00 60.67

 63 10 3.33 64.00

 64 12 4.00 68.00

 65 12 4.00 72.00

 66 13 4.33 76.33

 67 15 5.00 81.33

 68 7 2.33 83.67

 69 6 2.00 85.67

 70 8 2.67 88.33

 71 5 1.67 90.00

 72 8 2.67 92.67

 73 5 1.67 94.33

 74 1 0.33 94.67

 75 3 1.00 95.67

 77 1 0.33 96.00

 78 1 0.33 96.33

 79 3 1.00 97.33

 80 3 1.00 98.33

 83 2 0.67 99.00

 86 2 0.67 99.67

 95 1 0.33 100.00

 Total 300 100.00

.

4. Please find some english certificate group that can assist you in preparing good english manuscript

Reply 2.4. Thank you for your comments on the language of our manuscript. We have addressed them by utilizing a professional English editing service (Editage) during this second round of revisions. Please find the certificate attached.

Response to comments on May 15th, 2024 (Editors, reviewer 1 & reviewer 2) – Round 1

The title of study

All: We have modified the title of study as “Validation of the COmprehensive Score for Financial Toxicity (COST) in Vietnamese Patients with Cancer”. (page 1, line 1-2)

Reviewer 1

Reviewer #1: The paper offers a comprehensive validation study of the Vietnamese version of the Comprehensive Score for Financial Toxicity (COST) instrument among cancer patients. While the methodology is well-described and the study design seems robust, there are formatting inconsistencies and minor issues that require attention before publication.

1. Throughout the manuscript, there are multiple variations of the term for the Vietnamese version of COST. It is crucial to use only one consistent term for clarity. Please ensure that the same term is used consistently across the entire manuscript.

Reply 1: We have corrected all the terms used for COST, Financial toxicity and others statistical analysis. 

2. The paragraph format in the statistical analysis section lacks consistency. It is suggested to remove the headings "Descriptive statistics" and "Internal consistency" to maintain coherence.

Reply 2: We have modified this part for consistency. (page 5-6, line 106-129)

Statistical analysis

Continuous variables were summarized using means, standard deviations (SD), median, and Interquartile range (IQR). Categorical variables were described using frequencies and percentages. 

The reliability of the scale was assessed using Cronbach's alpha and McDonald's omega coefficients. A coefficient of 0.7 or higher generally indicates good internal consistency [24].

Construct validity involved two steps including exploratory factor analysis (EFA) and confirmatory structure analysis (CFA). EFA identified underlying factors influencing questionnaire responses. The scree plot and varimax rotation determined the number of factors retained [25]. EFA ensured clear and reliable factors through several strategies: (1) Removing items with low loadings (< 0.4) for minimal contribution; (2) requiring factors to have at least 3 items to capture the underlying construct adequately; (3) Removing items with high cross-loadings (> 0.3) to minimize ambiguity and enhance factor distinction [26]. Bartlett's sphericity test and Kaiser-Meyer-Olkin (KMO) confirmed the suitability of EFA [27, 28]. Building on the initial exploration of the factor structure using EFA (two factors loading was selected), CFA was then employed to assess the validity of the identified factor structure [27, 29]. Several fit indices (Root Mean Square Error of Approximation (RMSEA), Goodness-of-Fit Index (GFI), Comparative Fit Index (CFI), and Adjusted Goodness-of-Fit Index (AGFI)) indicated a good model fit (p < 0.001). 

Convergent validity was assessed using the Pearson correlation coefficient between the COST score and HRQoL (EQ-5D and EQ VAS). Shapiro-Wilk W test (p-value < 0.001) indicated that the COST score were not normally distributed due to skewness. Therefore, nonparametric tests were used to analyze the differences in COST score by characteristics. The Mann-Whitney U test was used for comparisons between two groups, and the Kruskal-Wallis test was used for comparisons between three or more groups. Additionally, a multivariate quantile regression analysis (at the 50th percentile) was conducted to investigate factors associated with the COST score. All independent variables were included for adjustment in the model. The threshold of a statistically significant level was less than 0.05. 

3. Recheck the format of references and citations to ensure consistency and accuracy. Pay attention to punctuation, capitalization, and formatting style (e.g., APA, MLA) to adhere to journal guidelines.

Reply 3: We have checked and corrected all citations and references in compliance with journal requirement. 

4. The term "Exploratory factor analysis" is redundant as it has already been explained previously. Consider removing it for clarity and conciseness.

Reply 4: Thank you for your thorough review. We have removed duplicate information from the statistical analysis and revised the entire section accordingly. (see reply 2, page 6, line 110-129)

5. Additional Comments: Please double-check for any formatting errors or inconsistencies throughout the manuscript to maintain readability.

Reply 5: Thank you again for your suggestion. We have carefully reviewed the document and made substantial revisions.

Reviewer 2

Reviewer #2: The study “Validation of the Vietnamese Comprehensive Score for Financial Toxicity (COST) in Cancer Patients” was performed by Tran et al.

The paper is well-written and showed that the COmprehensive Score for financial Toxicity (COST) demonstrated good internal consistency and The Vietnamese COST is reliable and valid.

There were several unclear issues and limitations that need to be addressed:

6. For the principal component analysis, two factors explained more than 64% of the variance, that a low explanation. With this analysis, you can consider adding more factors to have more than 80% of the variance.

Reply 6: Thank you for raising a critical point regarding the validation steps. After carefully checking our data and re-analyzing it, we found that achieving over 80% variance requires at least five factors to load. (Page 12, line 162-168) 

Unfortunately, this is not ideal for our scale with only 11 items. We have addressed this limitation in the discussion section on page 10 to further explain our results. 

However, in our data analysis, achieving an explained variance of over 80% required five-factor loadings, which is not ideal for the COST scale with only 11 items. Our final two-factor model demonstrated acceptable fit indices consistent with the findings of Yu et al. in China [15]. These comparisons suggest that the COST is likely to measure distinct but related constructs relevant to FT in patients with cancer. (Page 18, line 233-237)

We have copied output from AMOS for reference 

Total Variance Explained

Component Initial Eigenvalues Extraction Sums of Squared Loadings Rotation Sums of Squared Loadings

 Total % of Variance Cumulative % Total % of Variance Cumulative % Total % of Variance Cumulative %

1 6.003 54.572 54.572 6.003 54.572 54.572 3.747 34.067 34.067

2 1.135 10.317 64.889 1.135 10.317 64.889 3.390 30.822 64.889

3 .724 6.582 71.471 

4 .645 5.863 77.334 

5 .566 5.145 82.479 

6 .486 4.415 86.894 

7 .382 3.472 90.366 

8 .346 3.148 93.514 

9 .297 2.704 96.218 

10 .269 2.443 98.661 

11 .147 1.339 100.000 

Extraction Method: Principal Component Analysis.

7. Why did the authors express the COST score by both Mean ± SD and Median? This should be based on the distribution of COST score in each group, normal or non-normal distribution. Please also verify the statistical analysis, which tests were used to analyze the differences of COST score: t-test, ANOVA, Kruskal-Wallis or Mann Whitney?

Reply 7: Thank you for identifying the issues in our statistical analysis. We have addressed these concerns by revising the statistical method and incorporating the median, IQR (interquartile range), and p-value into Table 1.

Statistical method (Page 6, line 122-129):

Convergent validity was assessed using the Pearson correlation coefficient between the COST score and HRQoL (EQ-5D and EQ VAS). Shapiro-Wilk W test (p-value < 0.001) indicated that the COST scores were not normally distributed due to skewness. Therefore, nonparametric tests were used to analyze the differences in COST score by characteristics. The Mann-Whitney U test was used for comparisons between two groups, and the Kruskal-Wallis test was used for comparisons between three or more groups. Additionally, a multivariate quantile regression analysis (at the 50th percentile) was conducted to investigate factors associated with the COST score. All independent variables were included for adjustment in the model. The threshold of a statistically significant level was less than 0.05.

Table 1:

Chracteristics N (%) COST score Mean±SD Median IQR (25th -75th ) p value*

All 300 15.1 (7.3) 13 11-18 

Age 0.0747

< 60 152 (50.7) 14.1 (6.2) 13 11-16.5 

≥ 60 148 (49.3) 16.2 (8.2) 14 11-20.5 

Mean±SD

(Max – Min) 58.5±11.9 

Sex 0.0121 

Men 208 (69.3) 15.8 (7.8) 14 11-20 

Women 92 (30.7) 13.4 (6.0) 13 9.5-17 

Education level 0.0030 

Less than secondary school 95 (31.7) 13.5 (6.1) 12 11-15 

From secondary school and above 205 (68.3) 15.8 (7.8) 14 11-20 

Living areas 0.0031 

Urban 226 (75.3) 14.3 (6.8) 13 11-17 

Rural 74 (24.7) 17.7 (8.3) 15 12-23 

Religious 0.288

No 248 (82.7) 15.2 (7.5) 13 11-18 

Yes 52 (17.3) 14.4 (6.4) 13 11-17.5 

Marital status 0.961

Married 277 (92.3) 15.1 (7.4) 13 11-18 

Single/Widow/Dervoce 23 (7.7) 14.7 (7.1) 14 9-19 

Occupation 0.0001

Stable income jobs 66 (22.0) 18.8 (9.2) 16.5 13-23 

Unstable income jobs 163 (54.3) 13.5 (6.5) 13 10-16 

No income 71 (23.7) 15.4 (6) 14 11-19 

Economic status 0.0070

Poor 19 (6.3) 11.5 (6.0) 11 8-13 

Near poor 24 (8.0) 13.1 (6.2) 12 11-15.5 

Average 257 (85.7) 15.6 (7.4) 13 11-19 

Health insurance benefits 0.5507

100% 153 (51.0) 15.4 (8.1) 13 11-19 

95% 36 (12.0) 15.9 (6.6) 14 11.5-20.5 

80% 111 (37.0) 14.4 (6.5) 13 11-17 

Type of cancers 0.0252

Oesophagus cancer 39 (13.0) 12.3 (6.0) 12 8-15 

Stomach cancer 15 (5.0) 13.3 (3.3) 13 13-16 

Colorectal cancer 30 (10.0) 15.9 (7.2) 13 11-22 

Liver cancer 32 (10.7) 15.4 (7.1) 15 10-18.5 

Biliary/pancreatic cancer 112 (37.3) 16.6 (8.3) 14 11-21 

Breast cancer 72 (24.0) 14.2 (6.8) 13 11-15 

Stages of cancer 0.343

I 6 (2.0) 12.3 (2.8) 11.5 11-14 

II 54 (18.0) 13.8 (7.8) 13 9-18 

III 110 (36.7) 15.2 (7.1) 13 11-16 

IV 130 (43.3) 15.7 (7.4) 13 11-20 

Treatment modality 0.0385

Surgery 4 (1.3) 18.5 (17.4) 12.5 8.5-28.5 

Surgery and chemotherapy 153 (51.0) 16.3 (8.0) 14 11-20 

Chemotherapy 83 (27.7) 13.2 (5.8) 13 10-16 

Radiotherapy 54 (18.0) 15 (6.3) 13 11-17 

Palliative care 6 (2.0) 10.5 (4.0) 10.5 8-14 

Alcohol use 0.670

Yes 175 (58.3) 14.9 (7.3) 13 11-17 

No 125 (41.7) 15.4 (7.4) 13 11-19 

Cigarette status 11-17 0.953

Yes 146 (48.7) 14.9 (6.8) 13 10-19 

No 154 (51.3) 15.3 (7.8) 13 11-17 

8. A linear regression should be performed to understand the contribution of independent variables to the COST score.

Reply 8: We appreciate your suggestion once again. To address the skewed distribution of the COST score, we performed a multivariate quantile regression model to identify its predictors. We focused on the 50th percentile (median) of the COST score in this regression analysis. All results are presented in Table 4 and the results section (Page 15, line 196-197). 

Table 4. A multivariate quantile regression analysis for predictors of COST score.

Factors Coefficient [95% CI] p-value

Age group (≥ 60 vs. <60) 2.50 -0.12 to 5.12 0.062

Sex (Women vs. Men) -5.96 -10.39 to -1.54 0.008

Education level (From secondary school and above vs.Less than secondary school) 0.39 -2.46 to 3.24 0.786

Residence areas (Rural vs. Urban ) -4.57 -7.41 to 1.73 0.002

Religious (Yes vs. No) -1.50 -4.74 to 1.74 0.363

Marital status (Yes vs. No) 1.11 -3.44 to 5.65 0.632

Occupation (vs. Stable income jobs) 

Unstable income jobs -5.39 -8.71 to -2.07 0.002

No income -4.64 -8.63 to -0.65 0.023

Family economic status (vs. Poor) 

Near poor 0.82 -5.58 to 7.23 0.801

Average 2.96 -2.1 to 8.03 0.250

Health insurance copayment (vs. 100%) 

95% -4.07 -8.35 to 0.2 0.062

80% -0.86 -3.49 to 1.78 0.522

Types of cancer (vs. Oesophagus cancer) 

Stomach cancer 1.29 -6.64 to 9.21 0.750

Colorectal cancer 2.68 -3.32 to 8.68 0.380

Liver cancer 3.00 -2.39 to 8.39 0.274

Biliary/pancreatic cancer 2.04 -2.77 to 6.84 0.405

Breast cancer 0.07 -5.66 to 5.8 0.980

Stages of cancer (vs. Stage I) 

II 4.18 -4.63 to 12.99 0.351

III 4.36 -4.19 to 12.91 0.317

IV 5.11 -3.52 to 13.74 0.245

Treatment modality (vs. Surgery) 

Surgery and chemotherapy 1.68 -8.87 to 12.23 0.754

Chemotherapy -0.25 -10.8 to 10.3 0.963

Radiotherapy 0.11 -10.85 to 11.06 0.985

Palliative care -5.54 -18.94 to 7.87 0.417

Alcohol use (No vs. Yes) 2.86 -0.64 to 6.36 0.109

Cigarette smoking status (No vs. Yes) 2.36 -0.98 to 5.69 0.165

CI, Confidence Interval

---

## [Editor Report · Decision Letter 2]

17 Jun 2024

Validation of the COmprehensive Score for Financial Toxicity (COST) in Vietnamese Patients with Cancer

PONE-D-24-12510R2

Dear Dr. Tran,

We’re pleased to inform you that your manuscript has been judged scientifically suitable for publication and will be formally accepted for publication once it meets all outstanding technical requirements.

Kind regards,

Le An Pham, Ph.D,MD

Academic Editor

PLOS ONE
---

## [Editor Report · Acceptance letter]

19 Jun 2024

PONE-D-24-12510R2 

PLOS ONE

Dear Dr. Tran, 

I'm pleased to inform you that your manuscript has been deemed suitable for publication in PLOS ONE. Congratulations! Your manuscript is now being handed over to our production team.

Kind regards, 

on behalf of

Professor Le An Pham 

Academic Editor

PLOS ONE